# Far-Field Plume Characterization of a 100-W Class Hall Thruster

**Thibault Hallouin [1,2,]\* and Stéphane Mazouffre [1]** 

[1] Centre National de la Recherche Scientifique, 45071 Orleans, France; stephane.mazouffre@cnrs-orleans.fr
[2] Exotrail, 91300 Massy, France
\* Correspondence: thibault.hallouin@cnrs-orleans.fr

**Abstract:** The 100 W-class ISCT100-v2 Hall Thruster (HT) has been characterized in terms of far-field plume properties. By means of a Faraday Cup and a Retarding Potential Analyzer, both the ion current density and the ion energy distribution function have been measured over a 180° circular arc for different operating points. Measurements are compared to far-field plume characterizations performed with higher power Hall thrusters. The ion current density profiles remain unchanged whatever the HT input power, although an asymptotic limit is observed in the core of the plume at high discharge voltages and anode mass flow rates. In like manner, the ion energy distribution functions reveal that most of the beam energy is concentrated in the core of the plume $[-40°; 40°]$. Moreover, the fraction of low energy ion populations increases at large angles, owing to charge exchange and elastic collisions. Distinct plume regions are identified; they remain similar to the one described for high-power HTs. An efficiency analysis is also performed in terms of current utilization, mass utilization, and voltage utilization. The anode efficiency appears to be essentially affected by a low voltage utilization, the latter originating from the large surface-to-volume ratio inherent to low-power HTs. Experimental results also show that the background pressure clearly affects the plume structure and content.

**Keywords:** Faraday cup; Hall thruster; ion energy distribution function; ion current; plume; retarding potential analyzer

## 1. Introduction

Over the last decade, the satellite miniaturization has opened up a new space market. Technologically simple, inexpensive, and flexible, nano- and micro-satellites (1–200 kg) are well suited for various low Earth orbit missions, communication, science, and observation [1,2]. Propulsion systems have understandably adapted to this new market, through miniaturization and reduction of operating power. Compared to its chemical counterpart, electric propulsion is appropriated to challenges and needs of the micro-satellite market. In addition to a long operation time, one of the major advantages of electric propulsion is a high attainable exhaust velocity [1]. As a matter of fact, liquid or solid chemical thrusters are limited by the energy per unit of mass stored in the propellant. Thus, the maximum achievable exhaust velocity cannot exceed 5.5 km·s$^{-1}$. The electric propulsion uses an external energy source to accelerate the propellant. By decoupling energy source and propellant, high exhaust velocity can be reached, only limited by the electrical power delivered by the embedded power unit. This creates considerable benefits due to the decrease in propellant mass fraction which results in cost reduction and larger payload mass, and opens up the way to space missions requiring high velocity increments like orbit transfer, satellite displacement, and interplanetary journeys. Among all electric propulsion devices, Hall Thrusters currently offers the largest Thrust-to-Power ratio and thrust density with a specific impulse in the 1000–2500 s range. To answer the needs of the

micro-satellite market, HT dimensions have to be reduced to sustain a high ionization rate, and thus to operate the discharge at low power.

Many studies have been performed over the past few years to determine the performances and efficiencies of low-power Hall thrusters [3–15]. It is well known that low-power HTs exhibit performances below mid- and high-power devices due to a large surface-to-volume ratio. It is therefore of interest to measure small thruster plume properties in terms of current density profile, total ion current, and ion energy content to verify whether the size also has an impact on plume characteristics. Such measurements are of relevance for establishing interrelations between plume properties and HT characteristic dimensions in order to improve scaling relations for low-power HTs. Similitudes and differences with high-power HTs can also be established that helps with improving numerical simulation outcomes. Finally, plasma plume interrogation is also key to standardizing electrostatic probe designs for plume diagnostics.

In this contribution, we present performances of Hall thrusters operating below 300 W discharge power. After an overview of existing low-power Hall thrusters, this work focuses on a 100 W-class HT far-field plume examination in terms of ion current and ion energy. In addition, this study also aims at investigating the influence of the residual gas pressure on the plasma plume properties of miniature Hall thrusters.

## 2. Hall Thruster

### 2.1. Basic Principle

Electric propulsion provides a large panel of technologies with various performances, thereby offering flexibility according to the needs of the mission. Electric propulsion devices can be divided into two distinct categories: low $I_{sp}$/ large thrust and large $I_{sp}$/ low thrust. The $I_{sp}$, standing for specific impulse, is the impulse delivered per unit of propellant consumed. Hall thrusters belong to the first group with a large Thrust-to-Power ratio (60 mN/kW), a high thrust density (20 N/m$^2$) and a moderate $I_{sp}$ ($\sim$1000–2000 s). It is suitable for missions requiring high thrust level, such as drag compensation of Low Earth Orbit (LEO) satellites and orbit transfer. A Hall thruster uses an electrostatic force to accelerate ions [2,16,17]. A radial magnetic field is generated by permanent magnets or magnetizing coils inside an annular dielectric channel. A plasma discharge is established between an anode located at the back of the channel and an external thermo-emissive cathode. The crossed electric and magnetic field traps electrons, which locally increases the electron density, hence a high ionization of the propellant gas injected in the discharge channel. The electron mobility is reduced in the region of large magnetic field magnitude, which leads to a well-localized high axial electric field that accelerates the ions.

### 2.2. Low-Power Hall Thruster Performances

Although the Hall thruster technology in the 1–10 kW range is mature, miniaturization and low-power operation are still challenging. Figure 1 shows an overview of low-power Hall thruster performances with xenon as fuel and with BN/ BN-SiO$_2$ wall materials. Thrust ($T$), anode specific impulse ($I_{spa}$) and anode efficiency ($\eta_a$) are plotted as a function of the discharge power, in the range of 50–300 W. Low-power HT performances were collected in references listed in Table 1. In general high thrust - low $I_{sp}$ operation of a HT is achieved with large propellant mass flow and a relatively low discharge voltage (200–250 V). Typically, at 150 W discharge power, a Hall thruster generates about 10 mN of thrust with a specific impulse of 1100 s and anode efficiency of $\sim$40%.

**Table 1.** Sources of thruster performances presented in Figure 1.

| Hall Thrusters | References |
|---|---|
| BHT-100 | Szabo 2017 [3] |
| BHT-200 | Azziz 2003 [4] |
| HT100D | Ducci 2013 [5] |
| 50W HT | Lee 2019 [6] |
| 100W HT | Watanabe 2019 [7] |
| ISCT100-v2 | Hallouin 2019 [18] |
| ISCT100-v3 | Unpublished |
| ISCT200 | Unpublished |
| ISCT200-MS | Grimaud 2018 [9] |
| KM-20M | Bugrova 2001 [10] |
| SPT-20M | Loyan 2007 [11] |
| SPT-30 | Jacobson 1998 [12] |
| SPT-50 | Manzella 1996 [13] |
| T40 | Friemann 2016 [14] |
| ExoMG$^{TM}$-*nano* | Gurciullo 2019 [15] |

In Figure 1, we can observe important dispersion in the performances, reflecting a lack of knowledge of scaling laws. Indeed, the "ideal" HT design for discharge channel sizes and magnetic field magnitude and topology [19,20] is unknown, especially for low-power HTs. Figure 1 highlights that performances decrease as the discharge power is lowered, which is valid whatever the HT sizes and power ranges, see [21,22] and references herein. The thrust varies linearly from 2 mN to about 20 mN. The anode $I_{sp}$ remains relatively constant around 1100 s in the 100–300 W input power range. The $I_{sp}$ decreases quickly below 100 W, as can be seen in Figure 1. Interestingly, assuming a perfectly collimated beam with only singly-charged ions, such an $I_{sp}$ level corresponds to a 80 V acceleration voltage. This low voltage images the relatively low efficiency of small HTs: most of the thrusters have an anode efficiency around 20–40% in the 100–300 W range. This reduction in performance observed for low-power HTs is identified as an increase of plasma/surface interactions, as the discharge channel surface-to-volume ratio increases in small thrusters. The latter phenomenon is also responsible for a fast erosion process of the thin ceramic discharge channel wall, thus reducing the thruster lifespan [23]. Examples of mid- and high-power Hall thruster performances are given in Table 2 for comparison purposes.

**Table 2.** Example of mid- and high-power Hall thruster performances, operated with xenon at nominal operating point.

| HT Name | $P_d$ [W] | $U_d$ [V] | $\dot{m}_a$ [sccm] | $T$ [mN] | $I_{spa}$ [s] | $\eta_a$ | References |
|---|---|---|---|---|---|---|---|
| P5 | 5000 | 500 | 104.6 | 214 | 2086 | 0.44 | Walker 2004 [24] |
| SPT-100 | 1350 | 300 | 51.2 | 85.7 | 1706 | 0.53 | Sankovic 1993 [25] |
| BHT-600 | 615 | 300 | 24.5 | 41 | 1706 | 0.56 | Nakles 2009 [26] |

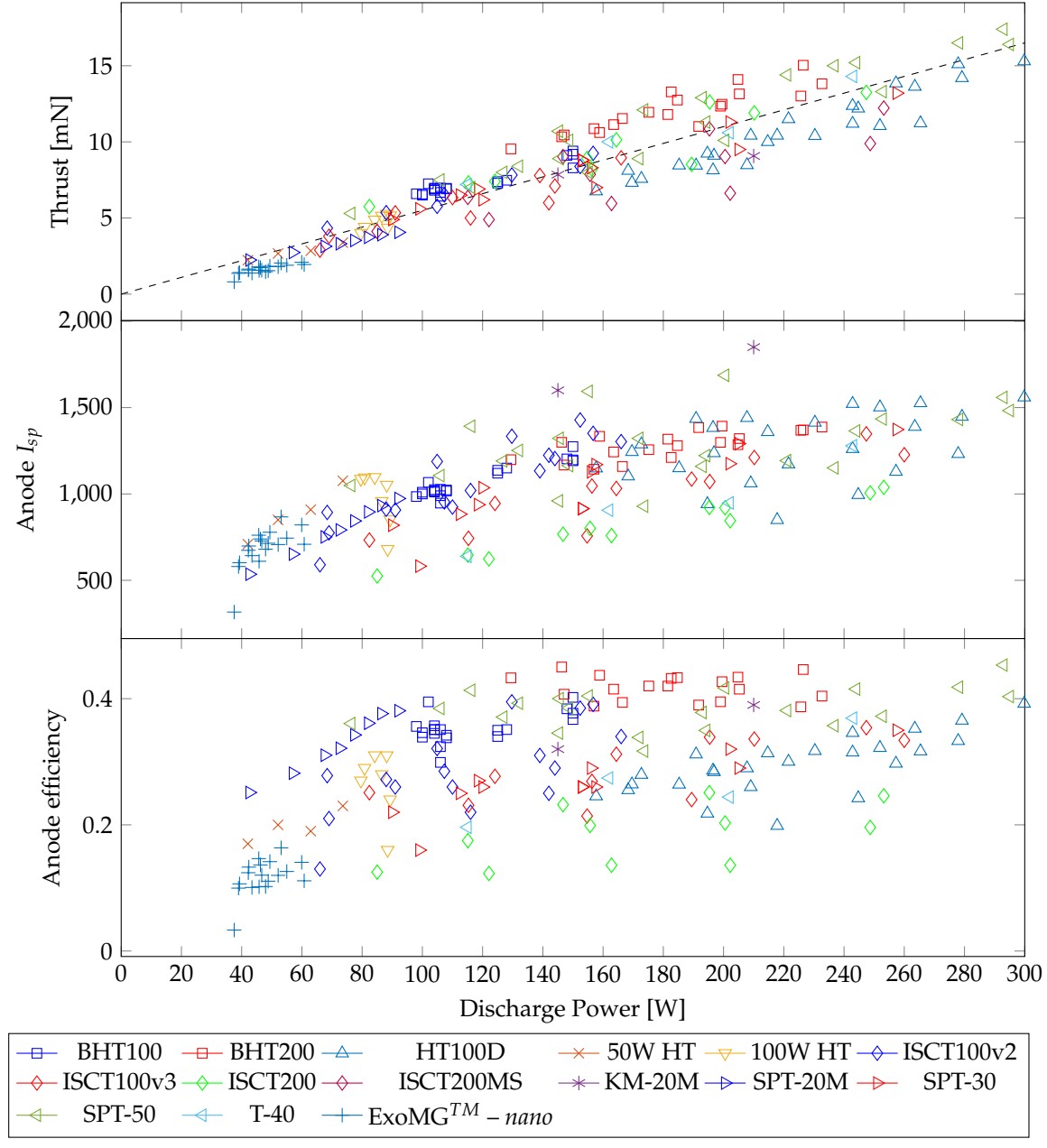

**Figure 1.** Thrust, anode specific impulse, and anode efficiency as a function of discharge power for low-power Hall thrusters, operated with Xenon and BN/BN-SiO₂ wall materials.

*2.3. 100 W-Class ISCT100-v2*

The ISCT100-v2, standing for ICARE Small Customizable Thruster, is a 100 W-class Hall thruster (see Figure 2), with performances comparable to the Busek BHT-100 [3]. The ISCT100-v2 corresponds to the $2S_0$-$2B_0$ configuration presented in Ref. 8 and constitutes an example of a low-power Hall thruster. The annular discharge channel is made of BN-SiO₂. A non-magnetic stainless-steel ring anode is placed at the back of the discharge channel, against the internal surface of the outer ceramic wall. The propellant gas is injected homogeneously inside the channel through a mullite disk, of which the high porosity allows for diffusion of the gas. The magnetic field is generated by means of cylindrical samarium-cobalt (Sm-Co) permanent magnets, located on both sides of the annular channel. The symmetrical distribution as well as the lense-shape of the magnetic field are provided by a pure iron magnetic circuit. The maximum magnetic amplitude is reached at the channel exit plane, while a near-zero amplitude is reached in the anode area.

Due to the use of permanent magnets, the magnetic field is assumed constant for all operating points, although an increase of magnet temperature, owing to an increase in discharge power, slightly modifies the magnetic field topology. For an increase of magnet temperature from 100 °C to 200 °C, the maximum radial magnetic field strength $B_{rmax}$ as well as the average magnetic field gradient reduce by 3% and 4%, respectively. The average magnetic field gradient is computed at the center of the discharge channel as the maximum radial magnetic field $B_{rmax}$ divided by the distance between the position of $B_{rmax}$ and the position of the zero magnetic field in the anode area [27].

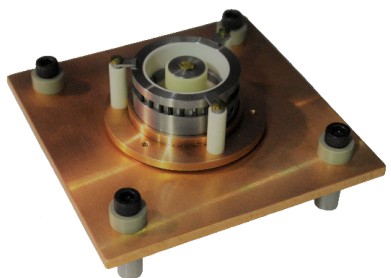

**Figure 2.** ISCT100-v2, 100 W-class HT.

A 1A-class LaB$_6$ hollow cathode was used to generate the electron current needed for discharge balance and beam neutralization. The cathode orifice is aligned with the thruster centerline and placed at 5 cm above the thruster inner pole and 1.5 cm upstream the channel exit plane. The inclination angle is fixed to approximately 45°.

During these experiments, high-purity xenon was used as the propellant gas for both the thruster discharge and the cathode discharge. The HT plume was investigated over a relatively broad discharge power envelope 70–200 W at different anode mass flow rates and discharge voltages, as presented in Table 3. Discharge current oscillations were recorded for each operating point with a Tektronix current probe (TCP202, DC to 50 MHz bandwidth). The probe was connected to a Tektronix digital oscilloscope (TDS5104, 1 GHz, 5 GS/s). The cathode was operated under steady conditions with 2 sccm of Xenon and a constant heating power adapted to maintain a Cathode Reference Potential (CRP) around −8 V.

**Table 3.** Tested operating points.

| $U_d$ [V] | $P_d$ [W] | $\dot{m}_a$ [sccm] |
|---|---|---|
| 200 | 72 | 5 |
| 300 | 132 | 5 |
| 200 | 92 | 6 |
| 300 | 159 | 6 |
| 350 | 203 | 6 |
| 200 | 110 | 7 |
| 300 | 201 | 7 |

## 3. Setup and Instruments

### 3.1. Vacuum Chamber

Plume diagnostic has been performed in the NExET—New Experiment on Electric Thrusters—vacuum chamber, a stainless-steel cylindrical tank 1.8 m in length and 0.8 m in diameter. NExET is fitted with a primary dry pump that evacuates 400 m$^3$·h$^{-1}$, and two turbomolecular pumps, with an overall pumping speed of 650 L/s-N$_2$. A cryogenic panel absorbs heavy-atoms such as xenon and krypton, the two main propellants used in electric propulsion. The pumping speed is around 8000 L/s when the 0.5 m$^2$ cold panel is sustained at 35 K. The overall pumping system guarantees a background operating pressure below $5 \cdot 10^{-5}$ mBar-N$_2$, proportional to the mass flow rate injected.

In the course of this study, the ISCT100-v2 was connected to a copper plate radiator, used to regulate the thruster body temperature, thus preventing magnets from overheating. The thruster body was kept at floating potential. The probes were mounted on an aluminum rotating arm, thin enough to limit plasma interaction in the vicinity of the measuring tools. The pivot point coincides with the exit plane of the thruster and is aligned with the thruster centerline as described in Figure 3. The probe centerline corresponds to the revolution axis of the thruster. The distance $R$ between the entrance aperture of the probe and the exit plane of the thruster was set at a distance of 31.5 cm, nearly 10 times the discharge channel diameter of the HT. The rotation motion is achieved by means of a Newport URS100BCC motorized rotation stage. The rotation angle $\theta$ is varied between $-90°$ and $90°$ with a step of $5°$, satisfying the condition $Rd\theta \geq d$ [28], with $d$ the orifice diameter of the probe. The thruster centerline corresponds to $\theta = 0°$.

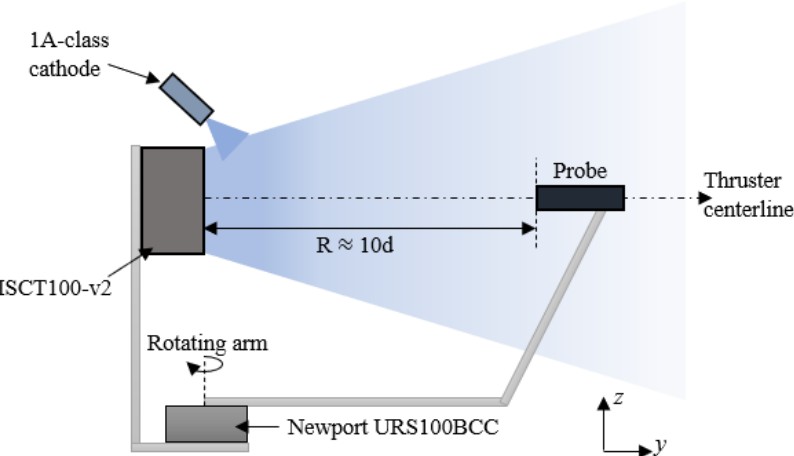

**Figure 3.** Schematic of the experimental set-up used for far-field plume measurements. The probe is aligned with the ISCT100-v2 centerline at the position $\theta = 0°$.

### 3.2. Faraday Cup

A Faraday Cup (FC) is an electrostatic planar probe with an isolated conductive cup. Compared to conventional planar probes, the FC is appropriate for accurately measuring the ion current density in the far-field plume of HT's [29,30]. The FC used in the present study, see Figure 4, is described in Ref. [29]. Numbers (1) and (6) refer to the FC aluminum body. The collimator (2) is made of graphite, selected for its low sputtering yield. Its orifice diameter is set to 10 mm to minimize charge exchange and scattering collisions within the cup due to gas pressure build up. Thus, the surface for current collection is 78.5 mm$^2$. The collector (5) is made of molybdenum to limit the secondary electron emission within the stainless-steel cylinder (4). The cylinder plus the collector form the cup. The collimator and the cup are electrically insulated using a PEEK spacer (3). A grounded and calibrated Keithley 2410 1100 V source-meter instrument has been used to measure the collected current. The unit was operated in voltage source to apply a constant negative bias voltage to the probe cup and to read the current. The bias voltage, namely the saturation voltage $V_s$, was set to $-50$ V. The influence of $V_s$ has been investigated and is described in depth in [18]. The collimator was set at floating potential. The current–voltage characteristic reveals that below $-50$ V the FC collects an almost constant current since plasma sheath expansion effect is drastically reduced. Faraday Cup measurements can be used to compute the total ion current $I_{i,tot}$ in the beam. Since $R \approx 10d$, the plasma

is assumed to originate from a source point. The total ion current, which is equal to the integral over an hemisphere of the ion current density $j_i$ therefore reads [28]:

$$I_{i,tot} = \int_{-\pi/2}^{\pi/2} \int_{-\pi/2}^{\pi/2} j_i(\theta,\phi)dS. \tag{1}$$

In spherical coordinates, the elementary surface $dS$ is defined by polar and azimuth angles, $\theta$ and $\phi$, respectively, such as: $dS = R^2 sin(\theta)d\theta d\phi$. The quantity $j_i(\theta,\phi)$ is not experimentally measured directly. Instead, only $j_i(\theta,0)$ is probed. We assume the plume properties are symmetrical and isotropic with respect to the thruster axis that is $j_i(\theta) = j_i(\phi)$. We obtain a new equation, which can be solved from FC data:

$$I_{i,tot} = \pi R^2 \int_{-\pi/2}^{\pi/2} j_i(\theta)sin(\theta)d\theta. \tag{2}$$

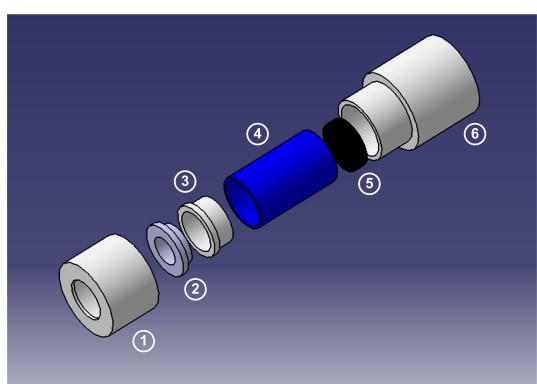

**Figure 4.** Exploded view of the Faraday Cup used in the present experiment, showing the main elements (see [29] for more details).

### 3.3. Retarding Potential Analyzer

A Retarding Potential Analyzer (RPA), also known as Retarding Field Electrostatic Analyzer (RFEA), is a gridded probe that uses electric fields acting as energy filters to selectively repel the constituents of a plasma or a beam [31–33]. The RPA used in this experiment is built with four electrostatically-biased grids and a collector (conductor) placed behind the grids that serves as a charge detector, as shown in Figure 5. All grids as well as the collector are aligned inside a stainless steel housing. The electrostatic grid assembly used to analyze the ion flux includes the entrance or screen grid (G1), the electron repeller (G2), the ion filter (G3), and the second electron repeller (G4), of which the role is effectively described in the literature [18,32]. An RPA acts as a high-pass filter: only ions with voltages that are energy-to-charge ratios, greater than the retarding grid voltage can pass and reach the collection electrode. The potential of the ion retarding grid is then varied while monitoring the ion current incident on the collector; thus, data are obtained as collector current versus discriminator voltage. A typical I–V curve is illustrated in Figure 6. Measurements were performed in the plasma plume of the ISCT100-v2 fired at 200 V with 0.5 mg/s xenon mass flow rate (Id = 0.36 A). The RPA was located on the thruster axis. The grid voltage scheme was: G1 floating, G2 at −15 V, G3 swept from 0 to 400 V, G4 at −20 V and collector at −5 V. The overall shape of the I–V curve depends upon the RPA design and the grid voltages. Deviation from the true (unperturbed) I–V trace is usually due to secondary electron emission and ionization and charge-exchange collision events inside the RPA [33]. The Semion control unit from *Impedans Ltd* has been used to power the RPA and acquire the I–V curves.

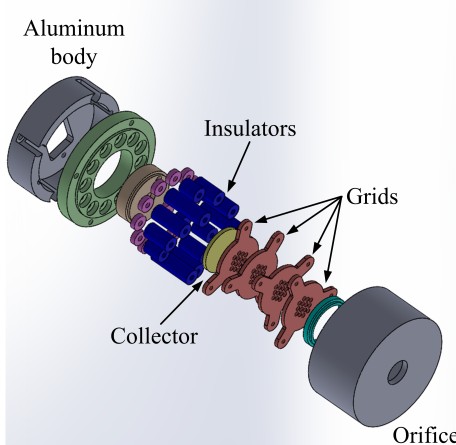

**Figure 5.** Exploded view of the 4-grid Retarding Potential Analyzer used in the present experiment.

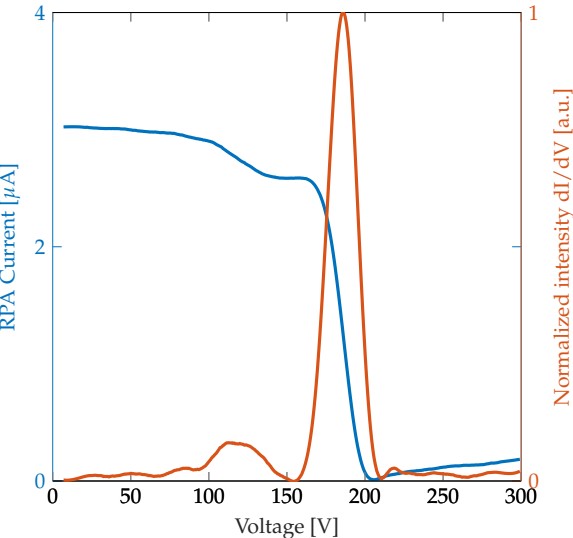

**Figure 6.** I–V characteristics (in blue) measured with the 4-grid RPA and the corresponding IEDF (in red) ($U_d$ = 200 V, $\dot{m}_a$ = 5 sccm, $\theta = 0°$).

The negative derivative of the I–V trace is proportional to the ion velocity distribution function [33]:

$$\frac{dI}{dV} \propto -f(v) \text{ for } v \geq 0, \tag{3}$$

where $I$ is the measured current, $V$ the discriminator voltage, $v$ is the ion velocity at the entrance of the RPA, and $f$ the ion Velocity Distribution Function (VDF). The previous expression is only valid for $v \geq 0$ as ions moving away from the probe ($v < 0$) never reach its orifice. Note that, in the case of a grounded aperture grid, a measurement of the plasma potential at the RPA location is needed to retrieve the true ion VDF. The velocity of ions entering the RPA is not the velocity in the plasma but the velocity behind the sheath instead. Assuming a planar sheath and for a collisionless medium, the kinetic energy of the ion is then increased by $eV_p$. In the course of this study, the entrance grid was floating and is supposed to be at the local plasma potential. However, an RPA does not measure the local ion VDF, whatever G1 voltage in fact, as all voltages have the ground as reference. Instead, it measures an accelerated VDF [31].

The distribution function obtained from the first derivative of the current is often called the Ion Energy Distribution Function (IEDF) when the velocity is converted into energy. However, the energy is then the energy of the ions along the direction of the RPA which is not necessarily the total kinetic energy. Only in the case of a perfectly collimated ion beam with a 0° divergence angle an RPA measures

the total kinetic energy. As all electric propulsion devices have a non-zero divergence angle, ions might have a significant amount of kinetic energy in directions perpendicular to the thruster axis.

The IEDF can provide two values used in the course of this study: the mean ion energy $E_{mean}$ computed from the first order moment of the IEDF and the most probable energy $E_{max}$ that corresponds to the energy of the highest peak.

The design of the 4-grids RPA used in this experiment has been developed ensuring a proper electric configuration to guide ions toward the collector. In that way, mesh size is of a few Debye length to adequately discriminate particles of plasma. Note that, for HTs, the average Debye length is between $10^{-2}$–$10^{-1}$ mm. As for the grid spacing, the gap must prevent from a build up of charge which would negate the effect of the previous biased grid. Thus, the standard space is approximately four times the Debye length [34]. Notwithstanding, sheath models show that grid spacing 100 times larger is still acceptable as, in Hall thruster cases, ions enter a RPA with a high velocity [35]. A collimator is placed upstream the screen grid to regulate the ion flux and define the probe view angle. Furthermore, material selection is essential to guarantee accurate measurements. PEEK grid mounts are used to avoid short circuits induced by material sputtering and conducting-coating formation. The collector is made of Molybdenum, a low secondary electron emission yield material, and grids in stainless steel. Characteristic dimensions of the 4-grids RPA operated during these experiments are given in Table 4.

**Table 4.** RPA—Characteristic dimensions.

| Characteristic dimensions | |
| --- | --- |
| Mesh size (mm) | 0.4 |
| Diameter (mm) | 45 |
| Length (mm) | 40 |
| Transparency (%) | 60.2 |
| Distance between grids (mm) | 2 |
| Collimator diameter (mm) | 10 |

## 4. FC Results

The overall shape of the $j_i$ angular profile remains similar in many features whatever the operating conditions, as shown in Figures 7 and 8. Indeed, the distribution is axisymmetric with respect to the thruster centerline ($\theta = 0°$) and reaches a maximum amplitude at this point. When moving away from the thruster centerline, the ion current density sharply decreases by an order of magnitude. Around $\pm 70°$, a hump is observed. This wing structure has already been noticed and identified as a backpressure effect [36]. The shape of $j_i$ profile for this 100 W-class HT is similar to the one observed with high-power HT plumes [37–39].

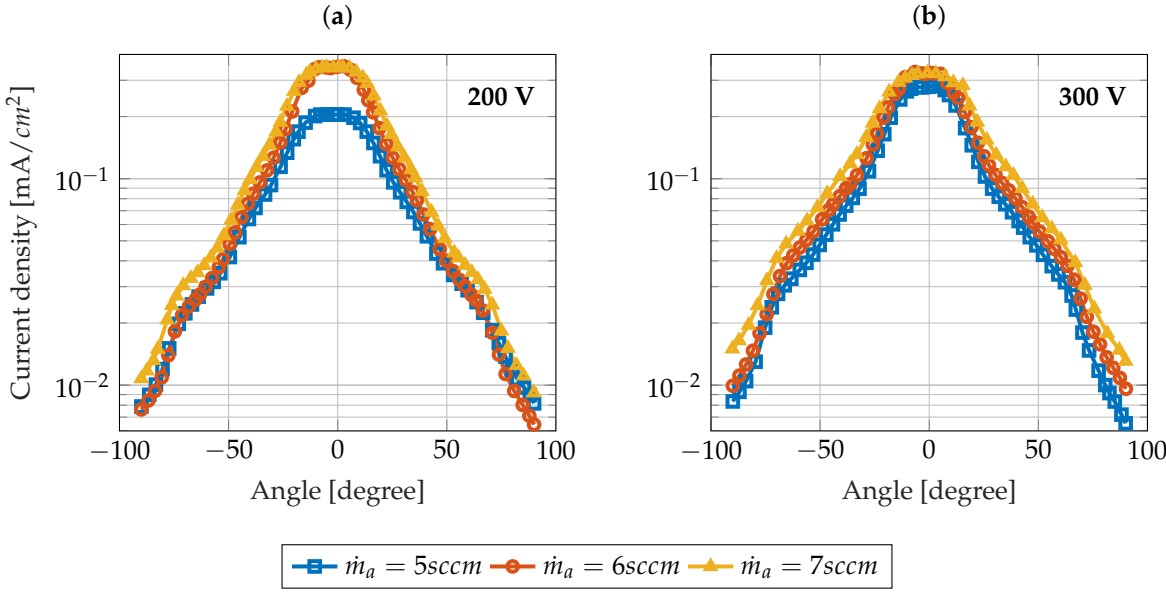

**Figure 7.** Semilog plot of $j_i$ angular distribution for different anode mass flows at (**a**) $U_d = 200$ V and (**b**) $U_d = 300$ V.

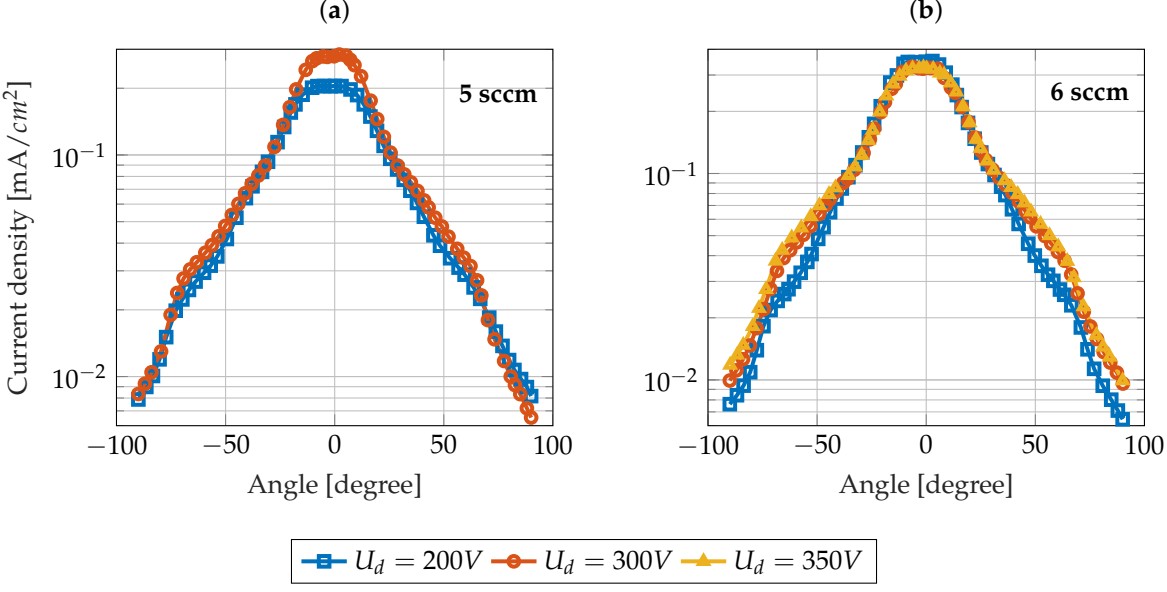

**Figure 8.** Semilog plot of $j_i$ angular distribution for different discharge voltages at (**a**) $\dot{m}_a = 5$ sccm and (**b**) $\dot{m}_a = 6$ sccm.

*4.1. Influence of the Anode Mass Flow Rate*

Figure 7a,b show the $j_i$ angular distribution at fixed discharge voltages, i.e., 200 V and 300 V, respectively, and different anode mass flows. As expected, the total ion current increases with the anode mass flow. As the neutral density increases in the discharge channel of the HT, ionization collisions become more likely, favoring production of ions. Nevertheless, interesting features are noticeable in the $j_i$ distribution. Firstly, the ion current distribution broadens when the mass flow rate increases. Secondly, the ion current density in the plume core (on axis region) reaches a limit value $\approx 0.35$ mA/cm$^2$ at both 6 sccm and 7 sccm.

The first feature is commonly observed in HT plume FC measurements and is identified as the consequence of an increase in vacuum chamber backpressure.

Numerous investigations have been performed to understand the impact of this parameter on ion current measurements. For instance, P5 (also called NASA-173M) [36,40], PPS-Flex [28], PPS$^{\circledR}$1350-G [41], SPT-100 [38,42], and BHT-1500 [39] plumes have been charaterized at different operating pressures and a fixed operating point. It appears that the core of the plume is unaffected by pressure changes in contrast to large angle parts, where the current density increases. The large difference observed in the plume large angle region is well understood and has been identified as the product of Charge EXchange (CEX) collisions between residual neutral atoms and fast ions. Indeed, if we consider a CEX collision occuring between a fast moving xenon ion $Xe^+_{fast}$ and a slow moving neutral $Xe^0_{slow}$, we obtain a fast atom $Xe^0_{fast}$ and a slow ion $Xe^+_{slow}$. In an HT plume, the radial potential gradient increases outward the on-axis region since the magnetic field lines have greater curvatures [43–47]. $Xe^+_{slow}$ resulting from scattering along the entire plume are then accelerated to the wings. As the backpressure increases with the mass flow, CEX events are more likely, which results in an increase of the current density in the wings. PPS-Flex plume characterization [28] also figured out that, above a pressure of $5 \times 10^{-5}$ mBar, the profile is distorted: the thruster is not operating in a standard mode due to backpressure effects. Thereupon, the backpressure condition is determinant in the reliability of ion current measurement. As the backpressure increases, the total ion current is overestimated, resulting in an overestimation of thruster performances.

The second feature can also be the consequence of backpressure effects. Since the CEX ions are redirected to large angles, the ion current density in the core of the plume is decreased. Nevertheless, other tendencies were described in literature [37,39] and lead us to consider further explanations, related to the inherent properties of low-power HTs. Previous studies have shown that operating mode are sensitive to operating parameters i.e., the discharge voltage, the anode mass flow rate and the magnetic field. Mode transitions in terms of current oscillations can occur [48,49]. Current oscillations are plotted as a function of discharge voltages (Figure 9a) and anode mass flow rates (Figure 9b) at constant anode mass flow rate and discharge voltage, respectively. The Standard Deviation (std) is computed to assess the oscillation amplitude at a given operating point. Standard deviation reaches about 87% and 85% when discharge voltages and anode mass flow rates are ramped up from 200 V to 350 V and from 5 sccm to 7 sccm, respectively. High-osccillation modes are produced at both high voltage and high anode mass flow rate, when the magnetic field topology is fixed. Such a high-oscillation mode affects thruster performances, namely the current utilization decreases and the plume divergence angle increases [48]. Consequently, the ion current density profile is attenuated in the core of the plume and broadened at larger angles, as observed in Figure 7a,b.

Note that the two ratios between the external discharge channel diameter $d_e$ and the vacuum chamber diameter $D$ and length $L$ were computed for this study and studies of Refs. [41,42]. Ratios are identical, namely $d_e/D \approx 0.04$ and $d_e/L \approx 0.01$–$0.02$.

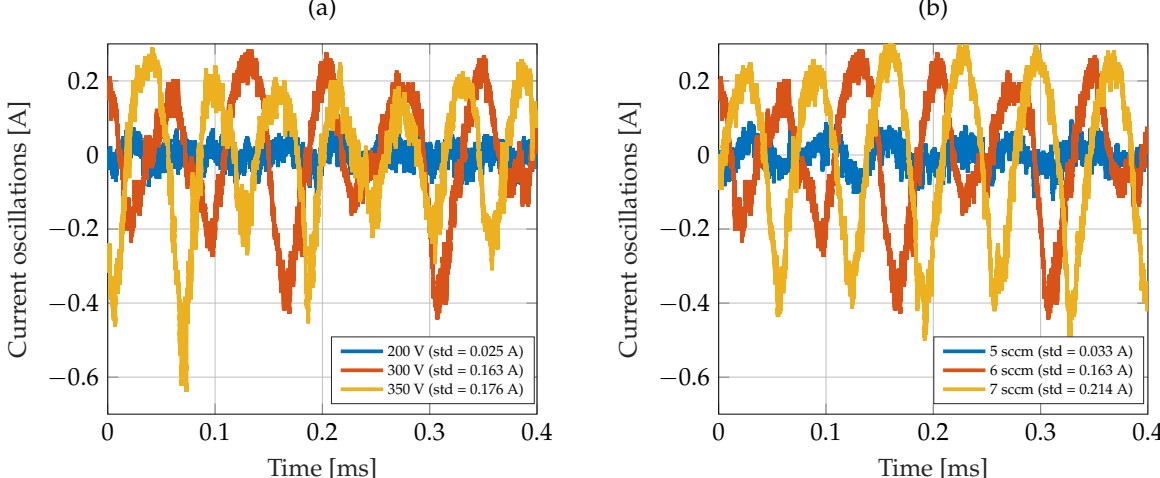

**Figure 9.** Discharge current oscillations at different operating parameters. (**a**) shows current oscillations for different discharge voltages at $\dot{m}_a = 6$ sccm and (**b**) shows current oscillations for different anode mass flow rates at $U_d = 300$ V. For each current oscillation, the standard deviation (std) is given. Oscillations are reduced to abscissa axis to have clearer comparison.

*4.2. Influence of the Discharge Voltage*

The influence of the thruster discharge voltage has been investigated for two different anode mass flow rates, namely 5 sccm and 6 sccm. The respective $j_i$ angular distributions are plotted in Figure 8a,b. Specific features are identifiable. Firstly, the ion current density increases at large angles when the discharge voltage is ramped up. Secondly, the current density increases in the core of the plume as $U_d$ is amplified, except for the 6 sccm case (Figure 8b), where the $j_i$ amplitude reaches a limit value whatever the discharge voltage.

The first observed feature diverges from common observations as usually an increase in discharge voltage leads to a narrowing of the profile and a growth of the central peak [37,39]. The discrepancy cannot be attributed to backpressure effects, since the gas injected through the anode remains constant and thus the residual pressure in the vacuum chamber is unchanged. The increased in ion current in the wings of the profile, i.e., when $\theta$ is above $\pm 40°$, and the resulting broadening when the discharge voltage is ramped up can have various origins. Firstly, as mentioned in several articles [50–52], high discharge voltage operation favors multiply charged ions production, of which the population increases at large angles. Secondly, the electric field distribution is also impacted by an increase in voltage, in such a way that the fraction of potential inside the thruster increases, shifting the location of the ionization area upstream and extending the acceleration region [53]. Some ions created at lower axial potential in the acceleration region are then accelerated toward large angles. In this work, the magnetic field stays unchanged. It is not increased when $U_d$ is increased.

The second reported feature is commonly observed in plume studies: an increase in discharge voltage favors ion production and the ion current increases in the core of the plume [37,39]. However, the specific case noticed in Figure 8b for 6sccm was rarely observed: D. H. Manzella and Sankovic J. M. mentioned this behaviour in the SPT-100 plume [38]. Although the asymptotic limit is consistent with high-oscillation operations (see Section 4.1), it can also be interpreted as a consequence of multiply-charged ion production. Such ions may be the result of collisions between high-energy electrons and CEX ions. Since the latter are slow moving (see Section 4.1), collisions with electrons are favored. Such ions are scattered to large angles which increases the ion current density at large angles, i.e., the divergence, and leaves the core unchanged. Moreover, multiply-charged ion production is favored at high voltage conditions, since both electron temperature [54,55] and ion density increase when the discharge voltage is ramped up. Further investigations need to be performed to evaluate the distribution of different ion species in the far-field plume.

**Table 5.** Ion beam current for each investigated operating point.

| $U_d$ [V] | $\dot{m}_a$ [sccm] | $I_d$ [A] | $P_d$ [W] | $I_{i,tot}$ [A] | $\eta_b$ | $\eta_m$ |
|---|---|---|---|---|---|---|
| 200 | 5 | 0.36 | 72 | 0.2805 | 0.779 | 0.764 |
| 300 | 5 | 0.44 | 132 | 0.3191 | 0.725 | 0.87 |
| 200 | 6 | 0.46 | 92 | 0.3499 | 0.761 | 0.795 |
| 300 | 6 | 0.53 | 159 | 0.3901 | 0.736 | 0.886 |
| 350 | 6 | 0.58 | 203 | 0.4097 | 0.706 | 0.93 |
| 200 | 7 | 0.55 | 110 | 0.4293 | 0.781 | 0.836 |
| 300 | 7 | 0.67 | 201 | 0.4769 | 0.712 | 0.928 |

Table 5 shows the total ion current computed for each operating point, using Equation (2). As expected, the beam current increases with both the discharge voltage and the anode mass flow. The current utilization and mass utilization efficiencies, termed $\eta_b$ and $\eta_m$ respectively, are also displayed in Table 5. They are respectively computed from Equations (4) and (5):

$$\eta_b = \frac{I_i}{I_d}, \tag{4}$$

$$\eta_m = \frac{\dot{m}_i}{\dot{m}_a} = \frac{mI_i}{e\dot{m}_a}, \tag{5}$$

$\dot{m}_i$ corresponds to the the singly-charged ions mass flow rate in the beam, $m$ the atomic mass and $e$ the elementary charge. $\eta_b$ is above 0.7 in average and $\eta_m$ above 0.8 which demonstrate a good ionization efficiency of the ISCT100-v2 despite the small size and a low discharge power. Similar efficiencies were described in literature for high-power HTs at the same discharge voltages [52,56,57]. However, it should be noted that $\eta_b$ decreases with discharge voltage, compared to what is observed for high-power HTs [56]. This difference can be interpreted as an increase of the fraction of electrons in the discharge current, i.e., an increase of electron leakage to the anode, owing to an increase of electron temperature with discharge voltage. While this loss can be mitigated in high-power HTs by increasing the magnetic field, the use of permanent magnets in low-power HTs prevents any correction. In fact, this effect could even be amplified as the magnets heat up when the power increases (see Section 2.3). High oscillation modes also favors the decrease of current utilization efficiency [48].

## 5. RPA Results

### 5.1. Ion Energy Distribution Function

The ion energy distribution function is investigated at low and high discharge voltages with 5 sccm of xenon. Two series of curves are shown in Figure 10a,b. Each IEDF is normalized to its maximum amplitude.

For the 200 V discharge voltage case, on the centerline of the thruster ($\theta = 0°$), a primary peak is visible at 186 eV that means 93% of the discharge voltage. The difference between these two values is in part the consequence of the CRP that is the energy required to extract electrons from the 1 A cathode. As previously mentioned, the CRP was maintained to $-8$ V in the course of this study, by adjusting the heating power. Other main loss terms are ionization, plasma–wall interactions, beam divergence, and overlap between the ionization and acceleration regions [53]. This primary peak can be identified as source ion population created by collisions between the magnetized electrons and the injected gas particle. A small population at 120 eV is also visible on-axis. These ions are probably created downstream of the mean ionization region. At 20° and 40°, the primary peak is still dominant and its energy drops a couple of volts, 181 eV and 175 eV, respectively. These two primary-ion populations are created further downstream of the ionization region, where the electrical potential is lower. Moreover, a large population of intermediate energy ions is also visible, of which the fraction increases with the angle. This ion population might be the result of different mechanisms. Firstly, source ions undergo

elastic collision resulting in the loss of momentum. Secondly, some ions are accelerated in regions of low plasma potentials where magnetic field lines have greater curvatures. These mechanisms result in low energetic ions, collected in the off-centerline region. From 60°, low-energy ions (57 eV) dominate the energy distribution. At this point, the primary population is still existing, but its fraction is low. Intermediate energy ions are also visible. Lastly, at 80°, the energy distribution is only dominated by low-energy ions (42 eV), as a result of CEX and diffusion collision events.

For the 300 V discharge voltage case, similar features are observed. Indeed, the centerline of the thruster is dominated by a high-energy peak of 286 eV, 95% of the applied discharge voltage. At 20° and 40°, the energy distribution is still dominated by source ions as well as intermediate energy ions. At 60°, the high-energy peak is still predominant compared to other energies. A large amount of low-energy ions also appears around 65 eV. At 80°, the dominant peak finally shifts toward low energies (52 eV). The source ion population is still visible (276 eV), yet its fraction is similar to the one of the intermediate energy ion population.

From previous observations, the angular IEDF can be divided into distinctive regions of which the angular boundaries vary with the operating voltage. The on-axis region is dominated by source ions that represent the majority of propellant ions. As the angle increases, intermediate energy ions are created as the result of electric potential distribution and elastic collisions. These two populations remain present until 60° and 80°, for the low and high discharge voltage cases, respectively. From 60°, low energy ion population remains important as a result of CEX collision events. Such regions were also founded in the far-field plume of high-power Hall thrusters, namely BHT-1500 [39,58], SPT-100 [42,52], and PPS®1350-G [41]. Although the angular boundaries change, the same ion populations are depicted, in accordance with far field-plume simulations [43].

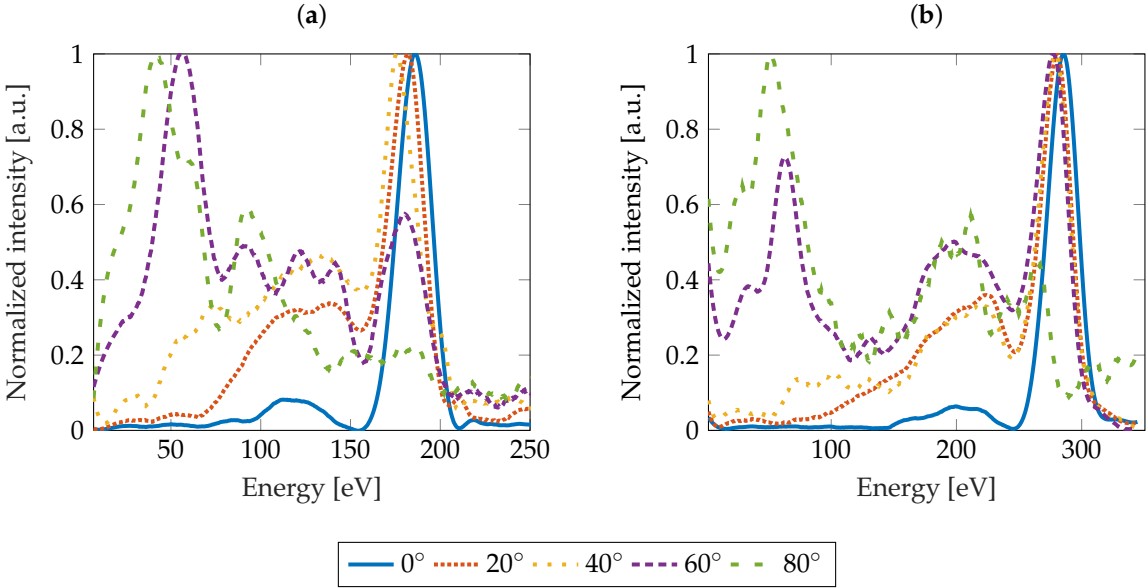

**Figure 10.** IEDF at different discharge voltages, 200 V (**a**) and 300 V (**b**), for constant anode mass flow rate (5 sccm). Curves are plotted for five different angles.

Figure 11 is a contour plot of the normalized IEDF as a function of the angle when the ISCT100-v2 operates at $U_d = 200$ V and $\dot{m}_a = 5$ sccm. A large fraction of ion flows within a narrow angle domain ($\theta \in [-40°; +40°]$) with a kinetic energy that approaches the potential energy provided by the thruster. At ±20°, tails are noticeable and reveal the presence of intermediate energy ions. At large angles, the associated current density of slow ions is relatively low and does not appear in this plot. The far-field plume is essentially dominated by source ions which create the useful net thrust.

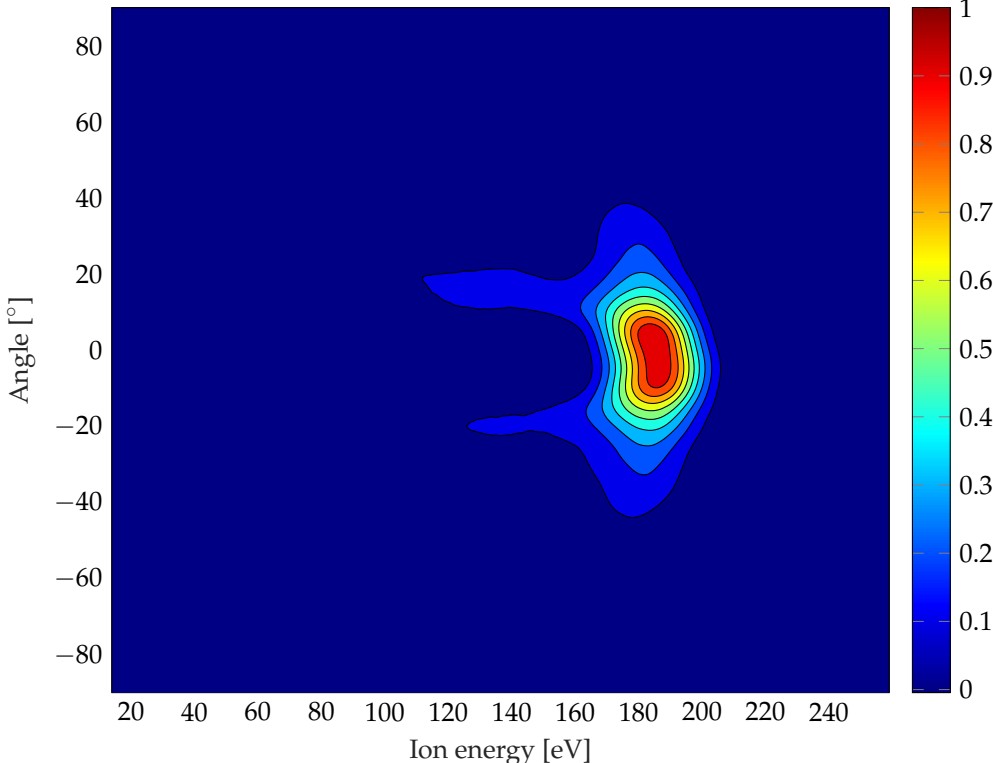

**Figure 11.** Countour plot of the normalized IEDF as a function of the angle (200 V, 5 sccm).

*5.2. Most Probable and Mean Energy Distribution*

The most probable and mean energy ($E_{max}$ and $E_{mean}$, respectively) can be determined from the IEDF. $E_{max}$ is the energy associated with the maximum of the IEDF and $E_{mean}$ is computed from the first order moment of the IEDF. The angular profiles of both $E_{max}$ and $E_{mean}$ is respectively plotted in Figure 12a,b. Regarding the $E_{max}$ distribution as a function of the angle, an abrupt drop at $\pm 60°$ and $\pm 80°$ is clearly identified, for the low and high discharge voltage case, respectively, in agreement with the IEDF description in the previous section. Furthermore, $E_{max}$ is maximum and remains constant for angles below $\pm 60°$ and $\pm 80°$, as can be seen in Figure 12a. This trend confirms that a large part of the plume is dominated by high-energy ions. However, the ion beam appears more focused in the low voltage case as the source ions are present within a narrower angular segment, which goes against common observations [39,41,42,52,58]. This difference is the consequence of difference in operating procedures. Whereas the magnetic field is usually increased with discharge voltage, it is kept constant in this experiment. Such conditions have an impact on the ion focusing mechanism [59,60].

The overall shape of the $E_{mean}$ angular distribution is not much affected by the discharge voltage as can be noticed in Figure 12b. The average energy is maximum on the thruster centerline, i.e., the core of the ion beam is dominated by source ions and not disturbed by scattering events. The average axial kinetic energy decreases steadily when moving off-axis as the low energy ion fraction increases with the plume angle. However, the decrease is not smooth as the angular distribution exhibits two humps, one at $\pm 30°$ and one at $\pm 70°$ as shown in Figure 12b. Humps mark the overlapping of two different ion populations. In addition, $\pm 30°$ is the overlapping between source ions and intermediate energy ions and $\pm 70°$ marks the border with the background gas environment, where the CEX ion fraction is dominant [43].

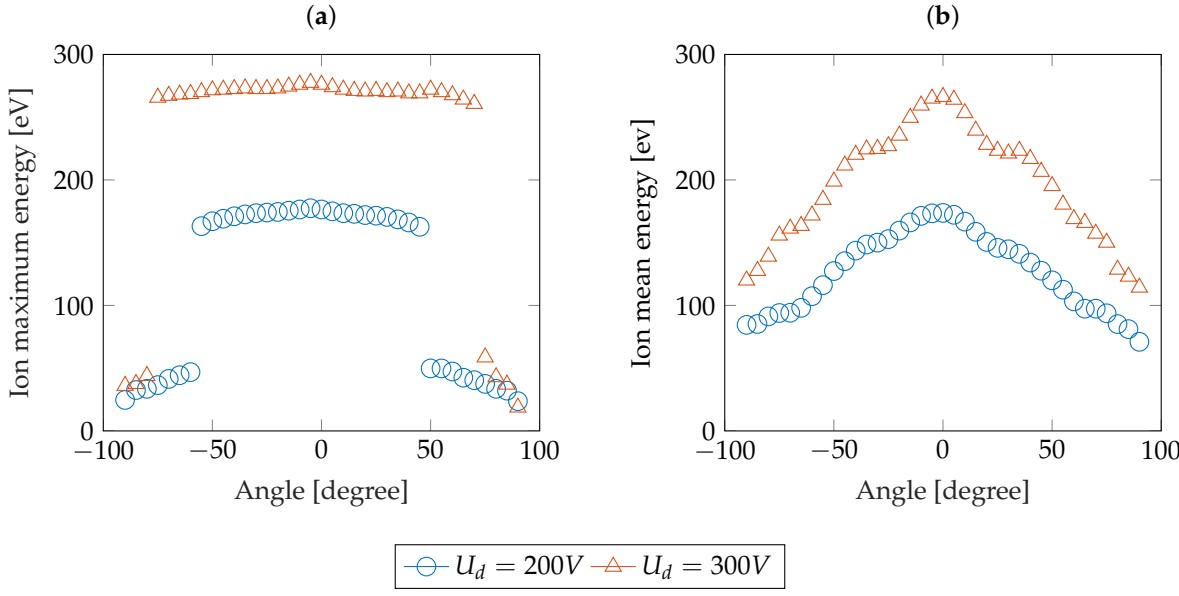

**Figure 12.** Angular distribution of the ion most probable energy (**a**) and mean energy (**b**), at 200 V and 300 V with 5 sccm mass flow rate.

Table 6 shows the beam voltage and the voltage utilization efficiency $\eta_v$ which is the ratio between $E_{beam}$ and $U_d$. The beam voltage is here defined as the mean energy in the beam. In other words, it is the mean of the mean energy angular profile. $\eta_v$ is constant around 0.7, whatever the operating conditions, unlike what is depicted in the literature for high-power HTs [56]. This difference is the consequence of loss mechanisms listed in Section 5.1, owing to a large surface-to-volume ratio of low-power HTs.

**Table 6.** Beam voltage for each investigated operating point.

| $U_d$ [V] | $\dot{m}_a$ [sccm] | $I_d$ [A] | $P_d$ [W] | $E_{beam}$ [eV] | $\eta_v$ |
|-----------|--------------------|-----------|-----------|-----------------|----------|
| 200 | 5 | 0.36 | 72 | 144 | 0.72 |
| 300 | 5 | 0.44 | 132 | 199 | 0.7 |
| 200 | 6 | 0.46 | 92 | 140 | 0.66 |
| 300 | 6 | 0.53 | 159 | 251 | 0.68 |
| 350 | 6 | 0.58 | 203 | 244 | 0.7 |
| 200 | 7 | 0.55 | 110 | 129 | 0.64 |
| 300 | 7 | 0.67 | 201 | 214 | 0.71 |

*5.3. Ion Flux*

Figure 13a,b show ion flux recorded with the RPA and FC as a function of angle from thruster centerline for 200 V and 300 V, respectively. Only half of the distribution is shown here for clarity. The RPA curves correspond to different filter bias voltage, i.e., to different energy lower limits. We term *RPA*0 the flux of ions without energy selection. Note that this flux is proprtional to the ion current measured by FC. Likewise, *RPA*20 represents the flux of ions of which the energy level is higher than 20 eV.

The shape of the *RPA*0 curve is similar to the profile probed with the FC: the amplitude is at a maximum at the thruster centerline and it decreases towards large angles. A hump is observed at $\pm70°$. The discrepancy in amplitude is attributed not only to operating conditions, as the two measurements were not performed simultaneously, but also to FC and RPA design differences.

As can be seen in Figure 13, all curves are similar below $-20°$ and the spread increases when moving to large angles. As the energy limit of the RPA increases, the distribution is narrower and

the hump is less marked. The variation of RPA curve shape gives insights into the various loss energy ion fractions and their role in the plume structure. It appears that a low energy ion fraction arises around $-30°$ and increases moving to large angles, as a result of CEX and diffusion events, see Section 5.1. The amplitude reduction of the hump at $-70°$ confirms that it is dominated by low energy ions resulting from CEX collisions, as mentioned in Section 4. Similar measurements were performed in the PPS®1350-G and SPT-100 far-field plumes [41,42] and the effect of background pressure was studied. It appeared that backpressure has a relevant effect on ion energy distribution. As the backpressure is decreased, the appearance of low energy ions is shifted to large angles and its fraction is reduced. Slow ion population identified in Figure 13a,b is then essentially the result of background pressure effects.

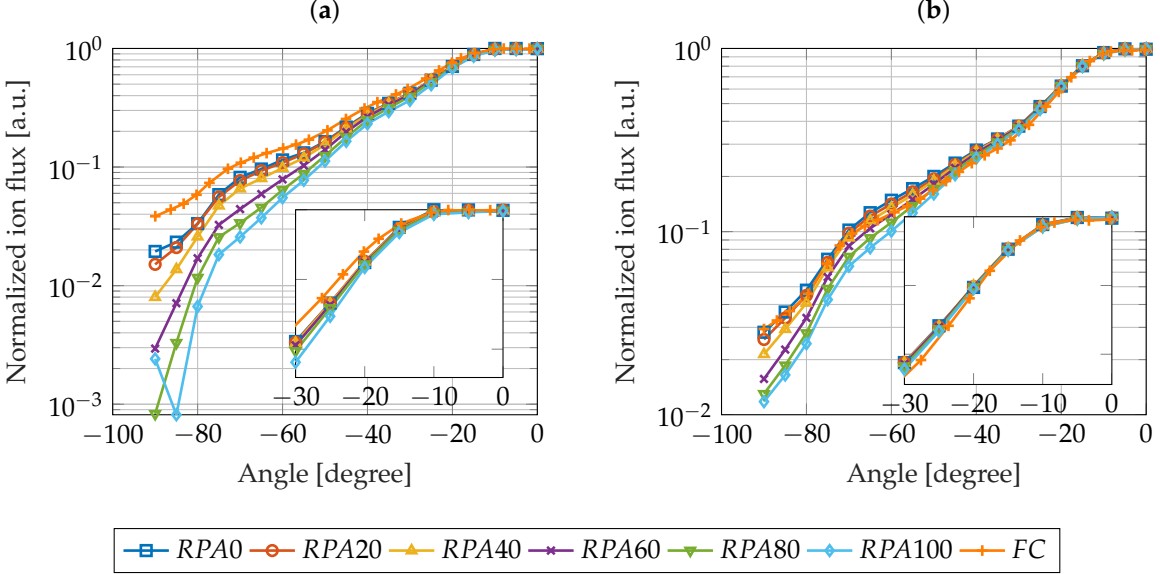

**Figure 13.** Normalized angular distribution of the ion flux recorded with a 4-grid RPA and the FC, at 200 V (**a**) and 300 V (**b**) applied voltages and a 5 sccm Xenon mass flow rate. The RPA curves correspond to different filter bias voltage, i.e., to different energy lower limits. The $[-30°; 0°]$ angular segment is shown in the insert for clarity's sake.

## 6. Conclusions

In this study, we have measured ion current density and ion energy in the far-field plume of the 100 W-class ISCT100-v2 HT, by means of a Faraday Cup and a 4-grid RPA, respectively. The impact of both anode mass flow rate and discharge voltage has been investigated and compared with far-field plume diagnostics of high-power HTs. It appears that the current density profiles remain similar whatever the HT input power, even though the 100 W-class HT is more sensitive to operating parameters. Indeed, in the core of the plume, the ion current density reaches an upper limit from a certain mass flow rate. The real cause is still misunderstood (inherent properties of the 100 W HT, background pressure effects) and calls for deeper experiments. In a similar way, the ion energy distribution function also reveals some interesting features. The ion energy is mainly concentrated in the core of the beam $[-40°; 40°]$ and the fraction of low energy ion populations increases at large angles, resulting from charge exchange and elastic collisions. Distinctive regions characterizing the arrangement of the plume are identified and remain similar to the one described for high-power HTs.

Furthermore, efficiency analysis has revealed interesting features. Both mass utilization and current efficiencies are consistent with the ones of high-power HTs, although low-power permanent magnets HTs suffer from a constant magnetic field when the anode voltage is ramped up. Nonetheless, voltage utilization appears more impacted by HT size. This difference is due to large surface-to-volume

ratio of low-power HTs. Thus, low anode efficiency of low-power HTs seems to essentially be the consequence of the lower fraction of anode voltage used in the acceleration process. However, further investigations of the effect of multiply-charged ions on performances are needed to supplement this work.

Moreover, background pressure appears to clearly affect the plume structure as it results in an increase of the fraction of low energy ions in the angular profile. Performing such measurements at lower operating pressure appears necessary to draw a line between ions created from inherent discharge mechanisms and those resulting from backpressure effects.

**Author Contributions:** Investigation, T.H.; Supervision, S.M.; Validation, S.M.; Writing—original draft, T.H. and S.M.; Writing—review and editing, T.H. All authors have read and agreed to the published version of the manuscript.

**Funding:** T.H. benefits from an Exotrail Ph.D. grant. This study was performed within the framework of the ORACLE joint-laboratory program. It has been partly supported by the Region Centre-Val de Loire council in the frame of the PEPSO-2 project.

**Acknowledgments:** The authors thank Thomas Gerard for his involvement in the experimental campaign and Antonio Gurciullo for his relevant advice.

**Conflicts of Interest:** The authors declare no conflict of interest.

## Abbreviations

The following abbreviations are used in this manuscript:

| | |
|---|---|
| CEX | Charge EXchange |
| FC | Faraday Cup |
| HT | Hall Thruster |
| IEDF | Ion Energy Distribution Function |
| ISCT | ICARE Small Customizable Thruster |
| LEO | Low Earth Orbit |
| NExET | New Experiment on Electric Thrusters |
| RPA | Retarding Potential Analyzer |
| std | Standard Deviation |
| VDF | Velocity Distribution Function |

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
