# Peer review of "Far-Field Plume Characterization of a 100-W Class Hall Thruster"

_aerospace, doi:10.3390/aerospace7050058_

Round 1

Reviewer 1 Report

Over all, the manuscript contains interesting information and a good point of comparison for future low-power Hall thruster research. However,the goal of this work and your main conclusions are not clear to the reader. Why do you want to characterize the plume of the ISCT100-v2? What specifically are you hoping to learn? Do you anticipate something about the plume will explain the low efficiency at low power?

Similarly, you ultimately conclude that the thruster's plume behavior is similar to higher-power thrusters. The main differences you suspect are due to keeping the magnetic field the same at all conditions. What does this mean for low-power thruster design? Did you find any clues about the origin of the low efficiency at low power based on the plume?

Aside from the motivation and conclusions of this work, a few specific points:

  • Could Fig. 1 be stated in terms of anode efficiency? The linear thrust vs power is not unexpected, so you really want to emphasize that the efficiency suffers at very low power. It might also be useful to include a few high-power points for reference.
  • What makes the ISCT100-v2 different from other low-power Hall thrusters? Is it expected to perform better than existing designs?
  • Figure 3 could be expanded to better describe the setup.
  • You mention that the RPA measurements are influenced by the plasma potential, and that you in fact measure "an accelerated VDF." Do you correct for this later? You discussion of this matter was somewhat unclear.
  • In Section 4.0.1 you suggest that "operating mode transitions" may affect the distribution. What do you mean by this? Do you have an evidence of this that you can present? It seems like a phenomenon worth investigating in more detail.
  • In Section 4.1 you suggest that increasing discharge voltage leads to a balance in increases Te and divergence of multiply-charged species such that a asymptotic current limit is reached in the plume core. Can you explain this further? Is there any quantitative analysis you can do about it?

Reviewer 2 Report

Major comments:

  1. Although the technical contents in this paper are nicely summarized and interesting, originality of this paper is not explicitly provided. The authors should explicitly show their originality either in Introduction or in Chapter 2.
  2. New findings should be explained in abstract; only objective and methodology are provided in the current abstract.

Minor comments:

  1. Definition of R on page 5 cannot be found.
  2. "discribe"  in line 162 on page 6 is misspelled and it should be corrected as "describe".
  3. "It is not increase ... " in line 257 on page 9 should be corrected as "It is not increased ...".

Reviewer 3 Report

This paper describes an experimental investigation of far-field plume characteristics of a 100-W class Hall thruster using a Faraday cup and retarding potential analyzer (RPA). The RPA data presented in Fig. 12 demonstrated that the amplitude reduction of the hump at -70° is dominated by low energy ions resulting from charge exchange collisions. The experimental results showed that background pressure appears to clearly affect the plume structure at the low-power Hall thruster in common with mid- and high-power Hall thrusters. This paper is informative experimental report for the development of low-power Hall thrusters and the improvement of Hall thruster’s scaling law. Therefore, in my own opinion, this paper deserves to be published in the Aerospace.

Round 2

Reviewer 1 Report

Thank you for the thorough revision of the manuscript. You addressed all of my previous concerns.